# Capacitive photocharging of gold nanorods

Felix Stete ⊕[1], Matias Bargheer ⊕[1,2] & Wouter Koopman ⊕[1] ✉

Light can charge plasmonic nanoparticles by photoredox reactions, significantly modifying their optical and chemical properties. However, the charging process has been challenging to track experimentally, severely hindering its thorough evaluation. In this study, we investigate the charging of gold nanorods during a light-induced reaction in situ, utilizing the sensitivity of the rods' longitudinal localized surface plasmon resonance to charge accumulation. Describing the particles as nanocapacitors, we present a model to quantify the number of charges on the particles and their connection to the illumination intensity. We find that the Fermi level, together with all other energy bands, is raised because of the repulsive potential of the additional charges. Experimental observations of the dependence on the solvent, the particle size, and ligand type further corroborate the proposed capacitor model. The results presented in this study lay the groundwork for the rational engineering of dynamic charge accumulation during plasmon-driven photoreactions.

Due to their broadband optical absorption resulting from localized surface plasmon resonances (LSPRs) and their ability to catalyze a broad variety of chemical reactions, noble metal nanoparticles are increasingly recognized as promising photocatalysts[1–3]. Both, the particles' characteristic optical and chemical properties are largely governed by the density of their conduction band electrons and are hence susceptible to the number of charges stored on the particle. Indeed, it was shown that charging the particle, e.g., by the application of an external voltage[4,5] or the addition of a strong reducing agent[6] alters both, the particles' plasmon resonance wavelength[6,7] and its ability to catalyze reactions[8].

With regard to their use as photocatalysts, a photoinduced chemical charging of noble metal nanoparticles is of particular importance. Charge accumulation as a result of photoassisted oxidation or reduction processes was reported for electrodes[9–11] and semiconductor nanoparticles[12,13] and is likely to play a role in photocatalysis[12–16] and photo-electrochemistry[9,10,17–19]. The relevance of photocharging for plasmon chemistry was first realized by Brus and co-workers who discovered a photovoltage present during the photoinduced growth of silver particles that they assigned to the light-induced oxidation of citrate[19–21]. Later, Jain and co-workers presented evidence for a photocharge-induced lowering of the activation energy for plasmon-driven reactions under illumination[22]. The rationale underlying this currently most advanced understanding of the

photocharging process can be summarized as follows: The decay of plasmon-excitations generates electron–hole pairs, either by interband excitation or by Landau damping[23]. These excited charges readily oxidize and/or reduce nearby molecules. If the half-reactions proceed with different rates, a dynamic excess charge accumulates on the particle. This photocharging is claimed to shift the Fermi level of the metal, thereby changing the activation barrier for charge transfer[22,24]. Such a mechanism might explain the photo-enhancement observed in several studies[19,20,25–30], including the reduction of $CO_2$[27,28], the oxidation of water[28], the anisotropic growth of gold nanoprisms[25], and the enhancement of multielectron reaction steps[30]. It could also be envisioned to selectively enhance particular plasmon-driven reaction pathways by systematically tuning the Fermi level through photocharging[28,29].

Despite the obviously huge potential for advancing plasmon-catalysis through photocharging, the current understanding of the photochemical charging process remains incomplete. In particular, the microkinetic picture described earlier does not consider the influence of the electric field generated by the charges themselves that counteract the charging process. Mutual repulsion of the charges and the associated raise in electrical potential might prevent the accumulation of more than a few electrons[31]. In contrast, the microkinetic model limits the accumulation of charge by the relative rates of oxidation and reduction. As a result, in the case that only one of the

[1]Institut für Physik & Astronomie, Universität Potsdam, Karl-Liebknecht-Str. 24-25, Potsdam 14476, Germany. [2]Helmholtz Zentrum Berlin, Albert-Einstein-Str. 15, 12489 Berlin, Germany. ✉e-mail: wouter.koopman@uni-potsdam.de

processes occurs, the model predicts the infinite accumulation of charge carriers on the particle. It is important to note that the authors of this model were aware of this shortcoming and proposed that Coulomb repulsion is probably a limiting factor to the achievable charge accumulation[22]. As they did, however, not make an attempt to include the Coulomb repulsion into their model, its impact on the charging process remains largely unclear. To achieve a deeper understanding of the nanoparticle photocharging process, elucidating this aspect is therefore crucial.

Currently, the presence of photocharging is mainly inferred indirectly from a reduced activation energy under illumination[22,26,29]. In such complex redox systems, it is, however, difficult to ascertain that photocharging is the only factor causing the reaction enhancement, unless an unambiguous probe of the charge on the nanoparticles is available. Photo-electrochemical measurements provide a more direct probe to charging effects[19–21], however, at the cost of an even higher complexity, which complicates the interpretation of the results and their extrapolation to purely photochemical systems. In particular, the involvement of electrodes and the presence of electrolytes strongly influence the capacitance of the system. Hence, an approach to study the photocharging of nanoparticles during a reaction would greatly advance the mechanistic understanding of the photochemical charging process.

For this work, we employed a more direct method to observe and discuss the light-induced charging behavior of solvated gold nanorods (AuNRs) in situ. The longitudinal LSPR of AuNRs is very sensitive to changes in the charge density[5–7], shifting to longer wavelengths (red-shift) for positive and to shorter wavelengths (blue-shift) for negative charging. Such charge-induced shifts were demonstrated by electrochemical charging[5,7] and by the addition of a strong reducing agent[6]. Here, we show that the resonance shift can also be induced by photoexcitation of the AuNRs. Monitoring this shift in situ confirms the presence of a significant photoinduced charge and allows us to identify several factors influencing the charging process, including light intensity, electron donor concentration, particle size, ligand type, and the solvent composition. The accumulated charge can be rationalized by a capacitor model, in which the driving potential is generated by interband charge separation. In contrast to earlier kinetic models, this approach intrinsically takes into account the fields generated by the charges. These fields induce the electrons to accumulate on the particle surface and are the cause of the elevation of the Fermi level. In fact, the entire band structure is raised due to the repulsive potential generated by the additional charges. The whole process is summarized in Fig. 1a. The results presented in this work represent a further step towards light-controlled "Fermi-level engineering" for the rational optimization of plasmon-driven catalysis.

## Results and discussion
### Photocharging of gold nanorods
To examine the accumulation of charges induced by light on nanorods, AuNRs were exposed to irradiation through a 532 nm laser beam in an oxygen-free ethanol/water mixture. At this wavelength, interband excitation generates high-energy holes in the $5d$-bands, which oxidize the dissolved ethanol (EtOH), transferring its electrons to the AuNRs. We carefully removed all electron scavengers, such as dissolved oxygen, from the solution prior to the excitation to stabilize the potentially induced charge. Moreover, to prevent unwanted side-reactions involving the ligand, the AuNRs were capped by polyvinylpyrrolidone (PVP), which is known to be stable against redox reactions over a wide potential window and has a high dielectric constant of $\varepsilon_{PVP} \approx 7.7$[32,33]. Moreover, PVP is known to leave sufficiently large pores, such that the EtOH molecules can reach the particles' surface[34].

As the plasmon resonance is highly sensitive to the charge density of the particle[5,6,35,36], we spectroscopically monitored the transmission of a colloidal solution to track the charging process. Figure 1b presents the evolution of the absorbance during 2.5 h of exposure to laser light with a power of 820 mW (circular excitation spot with diameter of 1 cm). At 700 nm, the longitudinal LSPR of the AuNRs is sufficiently far away from the excitation wavelength to allow monitoring alterations of the charge in situ. As expected for an increasing charge density, the longitudinal LSPR shifted towards shorter wavelengths (blue-shift). To track the shift in time, we determined the resonance from the center of a Gaussian fit to the longitudinal LSPR and plotted the change in resonance wavelength in Fig. 1c. Under illumination, the resonance steadily moves to shorter wavelengths and subsequently approaches a constant saturation value at longer illumination times. In the example shown here, the illumination induced a total shift of the LSPR of 4.5 nm.

The observed behavior is strongly reminiscent of charging a capacitor. Its saturation is then consistent with a saturating charge due to electrostatic repulsion[5]. However, charging is not the only possible origin of an LSPR shift. In particular, it can also be a result of a change in the rods' aspect ratio (e.g., by etching)[37] or in the permittivity of the environment[38], e.g., if the product has a different permittivity than the reactant. We therefore scrutinized the photocharging hypothesis by testing various routes for discharging.

### Oxidative discharging
Figure 2a shows the behavior of the LSPR after the laser is turned off. The particles are left in the dark for several hours. The resonance shifts back to longer wavelengths, but the shift saturates before the resonance reaches the original position it had prior to illumination. Assuming the initial blue shift originates from the nanoparticle charging, the LSPR's backshift must result from a partial oxidation-induced

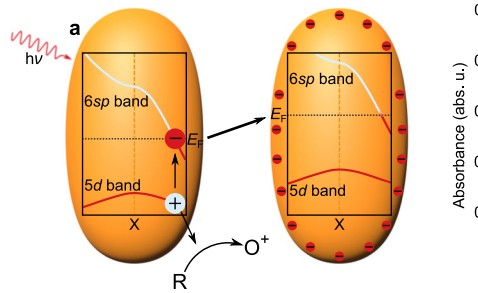
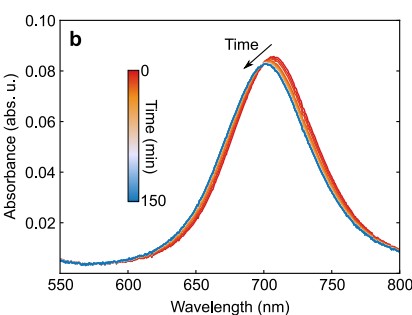
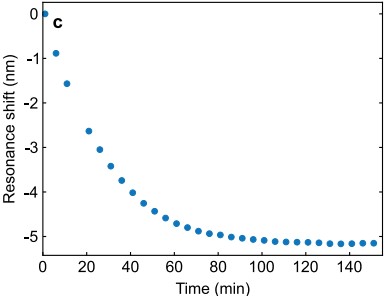

**Fig. 1 | Mechanism and light induced resonance shift. a** Absorption of a photon excites an electron--hole pair (here in the region of the X point in reciprocal space). The hole subsequently transfers to the surrounding hole scavengers, while the electron resides on the nanoparticle. Gradual transfer of charge increases electron density on the particle surface and, as a consequence raises the energy of the charges inside the particle. **b** Spectral evolution of the longitudinal LSPR during laser irradiation with 820 mW. The resonance blue-shifts due to the increasing number of charges on the particle. **c** Shift of the LSPR resonance maximum extracted from the spectra presented in (**b**). Source data are provided as a Source data file.

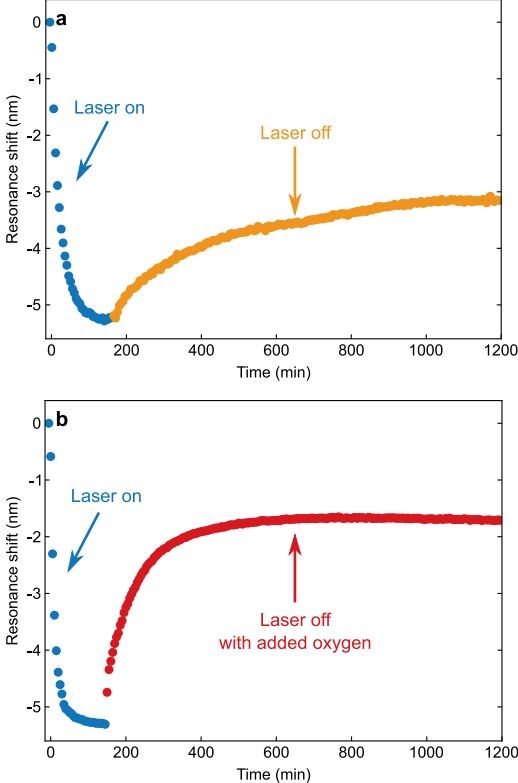

**Fig. 2 | Backshift due to discharging. a** Temporal evolution of the LSPR illuminated a laser power of 1080 mW (blue circles), including the backshift of the LSPR after turning off the laser and while keeping anaerobic conditions (orange circles). **b** Same experiment as (**a**), however, with oxygen added after turning off the light (red circles) by bubbling air into the solution. Source data are provided as a Source data file.

discharging of the rod. The most likely oxidizers are the protons at the interface, which were formed by the hole-driven oxidation of the alcohol[27]. Moreover, also residual trace amounts of oxygen, and the water itself can play a role[28]. In any case, the saturation of the shift below the initial wavelength indicates that the oxidants cannot remove the entire photoinduced charge.

Chemical changes, such as a modification of the AuNR geometry or of the environment's permittivit,y are expected to cause a permanent displacement of the resonance and are therefore probably not the origin of the observed shift. However, the permittivity in the vicinity of the particle might also be modified (electro)chemically. The backshift could then signify the diffusion of the reaction product away from the particle. Also, gold ions that were dissolved under radiation might re-adsorb as soon as the laser is turned off[24]. In order to exclude these processes, we centrifuged the particles immediately after illumination (6700 × g, 30 min) and replaced the supernatant solution with water. After washing, the particles initially retained the blue shift they had acquired during illumination and subsequently showed the same backshift towards the original resonance position as the unwashed particles (see Supplementary Note 1).

To further corroborate the observed LSPR shift as a result of photoinduced charge, we purposely added oxygen to the solution. This mediates the discharging of the particles by oxidation, which is expected to result in a backshift of the LSPR towards the original resonance wavelength. For the first 45 min after the laser was switched off, we bubbled the solution with air during the ≈2 min between each spectrum. After those 45 min, the cuvette was left with an open lid. Figure 2b confirms the much larger shift of the resonance back towards its original position compared to the quasi-anaerobic

conditions from Fig. 2a. Adding a strong oxidant thus removes almost all surplus electrons from the charged particles.

The slow discharging, on the order of minutes, and the presence of a residual charge are consistent with earlier reports on the impact of the photocharge on reaction kinetics. Lyu et al.[29] found that dark reactions involving palladium nanoparticles proceed faster if the particles had been illuminated in the presence of hole scavengers before they were added to the reaction mixture. Our discharging measurements confirm that the presence of a residual photocharge is indeed a realistic assumption.

In summary, the presented measurements strongly suggest that the shift of the longitudinal LSPR of AuNRs under illumination is caused by photocharging and can be used to as a proxy to study different influences on the charging process.

## Shift of the plasmon resonance by a surface charge

In the following, we will discuss how additional charge on the particle and the longitudinal plasmon resonance are connected. Previous reports explain the shift of the resonance $\omega_{LSPR}$ as consequence of an increased charge density $n$ in the bulk of the particle. As a result, they predict a proportionality between the LSPR and the square root of the total charge density: $\omega_{LSPR} \propto \sqrt{n}$[6,7,39,40]. However, to minimize the mutual repulsion, any excess charge on a small metal particle must accumulate at its surface. This insight results from both classical findings[41,42] as well as from more elaborate quantum mechanical calculations[5,31,36]. Thus, the bulk charge density remains unchanged. To be able to use the shift $\Delta\omega$ as indicator for the excess charge, it is therefore necessary to investigate the relationship between $\omega_{LSPR}$ and an excess surface charge density.

The Mie scattering problem for a bare sphere with a surface charge in a non-polar medium was solved by Bohren and Hunt in 1977[35]. It has recently been shown that this classical approach can also be used to determine the resonance position in charged gold nanorods in vacuum[36]. Importantly, in case of small shifts, this approach (as also the other mentioned models) shows a proportionality between the relative change in resonance $\Delta\omega/\omega_0$ and the relative change in the number of electrons on the particle $\Delta N_e/N_{e,0}$.

In the present investigation, the existence of a chemical environment comprising the ligand molecules and the solution results in a system that is considerably more complex than a simple metal sphere in a vacuum. It is thus not clear whether the proportionality between LSPR shift and induced charge holds. To evaluate the relationship between the particle's surface charge and its plasmon resonance freqeuncy, we adopted an experimental approach, utilizing the reducing agent thionine to assess the amount of charge stored on a gold nanorod after photocharging[29]. In its neutral form, thionine exhibits a pronounced absorption maximum. Upon transfer of two electrons, it is reduced to a colorless ion (thionine + 2e⁻ → thionine²⁻). We used this bleaching of the visible absorption to probe the amount of charge accumulated on a nanorod by UV-VIS spectroscopy. After photocharging the particles for 20 min under 1350 mW illumination power, 5 μL of 5 mM thionine solution was added to the dispersion of the charged particles. The spectral evolution after the addition of the thionine solution is shown in Fig. 3a. A notable decrease in absorption around 580 nm indicates the reduction of thionine, while the red shift of longitudinal LSPR indicates the discharge of the particle.

To assess the number of electrons that were transferred from the gold particles to the thionine molecules, we need to quantify the concentration of (unionized) thionine at each time a spectrum was recorded. To this end, we subtracted the spectrum of pure thionine (grey dashed line in Fig. 3a) multiplied with a concentration factor $\gamma$. The factor $\gamma$ was chosen such that the resulting difference spectrum reproduced the spectrum of pure AuNRs (grey dotted line in Fig. 3a) in the region between 550 and 600 nm. This region is not affected by the

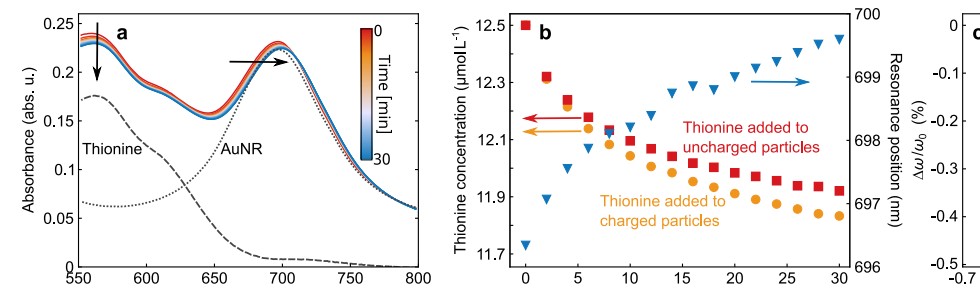

**Fig. 3 | Quantifying the charge by thionine reduction. a** Spectra recorded after adding thionine solution. The grey dotted line represents the AuNR spectrum right before adding thionine. The grey dashed line represents the thionine spectrum from which the thionine concentration is determined. **b** Thionine concentration after addition to the charged (red squares) and uncharged (orange diamonds) particles plotted together with the plasmon resonance position (blue triangles). **c** Relative shift in resonance plotted over the relative change in electron numbers on one particle as determined from thionine reduction (blue circles) and a linear fit to this data (orange line). Source data are provided as a Source data file.

plasmon shift and can therefore be used as reference. To account for the influences of the colloidal AuNR dispersion on the thionine spectrum, we determined the "pure" thionine spectrum by subtracting the thionine-free AuNR spectrum recorded right before from the first spectrum right after the addition of thionine.

Multiplication of the factor $\gamma$ with the original concentration of 12.5 µM yields the thionine concentration at any time after the addition to the charged particles. The resonance position of the longitudinal plasmon can be determined from each spectrum after subtraction of the thionine contribution. Figure 3b shows the temporal evolution of the thinonine concentration (orange circles), as well as the shift of the longitudinal plasmon resonance (blue squares). However, the recorded decrease in the concentration of thionine does not directly represent the amount of charge extracted from the particles, since the addition of thionine to uncharged particles also resulted in a bleaching of the dye (see the red squares in Fig. 3b), albeit to a lesser extent. Because this bleaching cannot be related to the transfer of photo-induced charges from the nanorods, we utilized the difference between the two concentrations to measure the number of electrons that are transferred during the discharge of the particles. Using the known number of gold particles in the solution ($1.5 \cdot 10^{11}$ mL$^{-1}$), we then obtained the number of electrons per nanorod $\Delta N_e$. Figure 3c finally presents the relation between $\Delta \omega / \omega_0$ and $\Delta N_e / N_{e,0}$. The results clearly show the linear relationship between the two values, with a slope of $\Delta \omega / \omega_0 = 0.7 \Delta N_e / N_{e,0}$. In the following sections, we will use this proportionality to relate the measured spectra to the photoinduced charge.

**Intensity dependence of charging: capacitor model**

To gain a deeper insight into the mechanism behind the photo-induced charging, we exploited our direct access to the charge state to study its dependence on the illumination intensity. Figure 4a presents the temporal evolution of the LSPR shift with respect to its initial value, $\Delta \omega = \omega - \omega_0$, for various laser powers ranging from 100 to 1620 mW. All measurements show qualitatively the same behavior observed before, consisting of a steep initial blue-shift of the LSPR, followed by a saturation at longer times. Clearly, with increasing illumination power, the resonance moves faster and reaches higher energies before the shift saturates. In the following, we will show that this behavior can be understood by regarding the particles as electrostatic nanocapacitors.

In the picture of a nanoscale capacitor, a photovoltage $U_{photo}$ causes a flow of electrons from the solution into the particle. This process eventually builds up an excess surface charge $q_e \Delta N_e$. When the potential of the added charge fully compensates $U_{photo}$, the charge transfer ceases. Treating the AuNRs as capacitors, the maximum additional charge, $q_e \Delta N_{e,max}$, that can be stored is determined by the particles' capacitance:

$$C = -q_e \Delta N_{e,max} / U_{photo}. \tag{1}$$

The temporal evolution of the charge on the particle is given by the equation for the charging of a capacitor:

$$N_e = N_{e,0} + \Delta N_{e,max}(1 - e^{-\frac{t}{\tau}}). \tag{2}$$

Here, $N_{e,0}$ denotes the bulk charge before the onset of the illumination and $t$ the time after the illumination started.

The relative resonance shifts in our experiments are below 1%. We can therefore use the proportionality between $\Delta \omega / \omega_0$ and $\Delta N_e / N_{e,0}$ for small changes (the value of 0.7 of the proportionality constant, deduced previously, is in principle not of importance here as it cancels out) and directly express the evolution of the resonance position as

$$\omega = \omega_0 + \Delta \omega_{max}(1 - e^{-\frac{t}{\tau}}). \tag{3}$$

Here, $\Delta \omega_{max}$ describes the maximum shift that is induced by the maximum charge on the particle. The respective fits, represented as solid lines in Fig. 4a demonstrate that indeed, this approach accurately describes the temporal evolution of the LSPR.

In the following, we take a closer look at the two parameters that characterize the charging process: The charging time constant $\tau$ and $\Delta N_{e,max}$, the additional charge stored on the particle under illumination for $t \rightarrow \infty$. Figure 4b presents the dependence on the illumination intensity of the relative resonance change $\Delta \omega / \omega_0$, which directly translates to the relative charging $\Delta N_{e,max} / N_{e,0}$, in our case by dividing by 0.7. At the maximum measured power (1620 mW), the relative increase in the amount of charge is $\Delta N_e / N_{e,0} \approx 1.33\%$.

A series of previous publications established that the accumulation of electrons on gold NPs results from the transfer of excited $d$-band holes to a hole scavenger[22,29]. Without optical excitation, the chemical potential of holes equals the chemical potential of free conduction band electrons, $\mu_e$, which is identical to the Fermi level, $E_F$. Under illumination, interband excitation generates holes in the $5d$-band, which have a much higher oxidation power compared to the conduction-band holes in thermodynamic equilibrium. The photo-voltage $U_{photo}$ generated by the excitation thus corresponds to the potential drop between the electrochemical potential of photoexcited holes, $\mu_h^*$, and the Fermi level of the particle, $E_F$.

$$U_{photo} = (\mu_h^* - E_F)/q_e \tag{4}$$

In Eq. (4), the dependence on the incident light intensity is included in $\mu_h^*$, while we assume that the number of excited $sp$-band electrons is

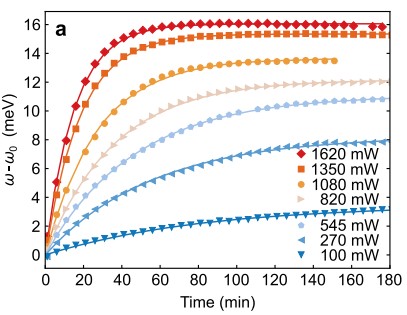
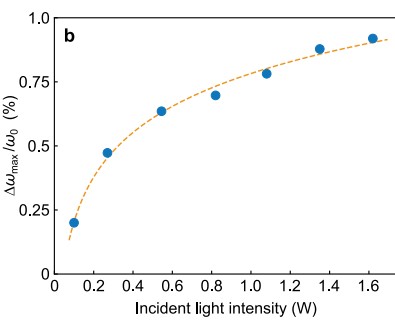
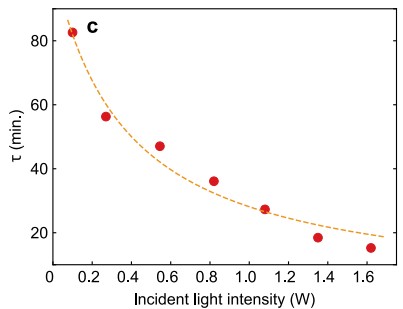

**Fig. 4 | Power dependent charging. a** Resonance positions during illumination with various laser powers between 100 and 1620 mW (colours and markers as depicted in the legend). Values for $\Delta\omega/\omega_0$ (**b**, blue circles) and $\tau$ (**c**, red circles) were extracted from the data in (**a**) via Eq. (2) (best-fits are presented as solid lines in (**a**). The dashed orange lines represent the logarithmic best-fit to $\Delta\omega/\omega_0$ (b) according to Eq. (6) and the exponential best-fit to $\tau$ (c) according to Eq. (7) describing intensity dependent photocharging of a nanocapacitor. Source data are provided as a Source data file.

sufficiently low, such that $E_F$ remains constant. This voltage can be regarded as the external voltage that is applied to a capacitor. The potential that drives the holes to leave the particles is then given by the difference between $\mu_h^*$ and $\mu_{sol}$. Similar to a classical capacitor, the charging process induces an additional potential $U_{charge} = E_F - \mu_{sol}$ that works in opposite direction of the external voltage. Charge is transferred between the particle and the environment until the two potentials $U_{photo}$ and $U_{charge}$ cancel each other out.

We now want to derive an intensity-dependent expression for $U_{photo}$. To express $\mu_h^*$, we borrow the concept of quasi-Fermi levels from semiconductor physics. The quasi-Fermi level $\mu_h^*$ quantifies the average occupation probability of holes in the $5d$-band. As long as $\mu_h^*$ is located between $\mu_e$ and the upper edge of the $d$-band, the distribution of holes in the $5d$-band follows Boltzmann statistics, and the quasi-Fermi level can be expressed as:[43,44]

$$\mu_h^* = E_{5d} + k_B T \cdot ln\left(\frac{N_{5d}}{n_h + n_h^*}\right). \qquad (5)$$

Here, $E_{5d}$ denotes the upper $d$-band-edge and $N_{5d}$ is the so-called effective density of states. Eq. (5) is valid under the condition that the Boltzmann distribution is an appropriate description of the distribution of excited holes in the valence band[43,44]. This is the case as long as the logarithmic expression is greater than one, implying that the chemical potential is more than $k_B T$ higher than $E_{5d}$. In the following, we use that the density of holes in the $5d$-band in thermal equilibrium, $n_h$, is much smaller than the density of excited electrons, $n_h^*$, $(n_h \ll n_h^*)$ such that $n_h^* \propto I_{abs}$. Combining Eqs. (1), (4), and (5), we arrive at an expression for the photoinduced charge:

$$\Delta N_{e,\,max} = \frac{C}{q_e^2} \cdot \left[E_F - E_{5d} + k_B T \cdot ln\left(\frac{n_h^*}{N_{5d}}\right)\right]. \qquad (6)$$

For clarity, the entire photo-charging process that is incorporated in Eq. (6) is illustrated in Fig. 5. The absorption of photons excites interband transitions, either by direct absorption or via plasmon decay[22]. The resulting electron–hole pairs generate a non-equilibrium situation, in which the chemical potential $\mu_h^*$ of the excited holes shifts towards the upper edge of the $5d$-band, as described by Eq. (5). Increasing the photon flux $I$, increases the hole density in the $5d$-band, $n_h^*$, and hence $\mu_h^*$ increasingly shifts towards the band edge. The shift of the potential is manifested by the photovoltage $U_{photo}$, and the difference between $\mu_h^*$ and $\mu_{sol}$ acts as driving force for the transfer of holes out of the particle. The additional electrons residing on the particle accumulate at its surface. Outside of the particle, the excess charge is screened by an electrical double layer that stabilizes the particle[45]. The charges generate an electrical potential difference

$U_{charge}$ between the particle and the environment that raises the entire band structure, including $\mu_h^*$. In doing so, $U_{charge}$ decreases the difference between $\mu_h^*$ and $\mu_{sol}$ up to the point where the two are equal, and the charge flow ceases.

As shown by the dashed orange line in Fig. 4b, Eq. (6) reproduces the dependence of $\Delta N_{e,\,max}/N_{e,\,0}$ on the light intensity very well. The best fit using the nanocapacitor model gives a capacity of $2.7 \cdot 10^{-15}$ F per particle or an areal capacitance of 294 μF cm$^{-2}$ (see "Methods"). Electrochemical landing experiments recently showed similar areal capacitance[46–48]. We therefore conclude that the capacity determined from our measurements is of plausible magnitude.

An important implication of the nanocapacitor model is that the change in the Fermi level of the particle is caused by the electrostatic potential of the (additional) surface charges. Therefore, the entire band structure is shifted to higher values, and the energy gap between the $5d$-band and the Fermi level remains constant. Consequently, the strength of the interband absorption is maintained during the charging process. The notion that the Fermi level is elevated due to the filling of the conduction band, whose energy stays constant, is therefore misleading.

The nanocapacitor model developed in Eqs. (1) to (6) differs from the microkinetic model established earlier to describe a similar situation[22]. In this previous model, the excited conduction band electrons and $5d$-band holes are treated as reactants in the reduction and oxidation processes, and the accumulation of charges results from an asymmetry in the reduction and oxidation rates of the nanoparticle. The model was used to describe a change in the Fermi level of gold nanoparticles due to charge accumulation during the parallel oxidation of EtOH and reduction of ferricyanide $\left([Fe(CN)_6]^{3-}\right)$. The resulting predictions of this microkinetic model deviate from those of the nanocapacitor model in several aspects. In contrast to the logarithmic dependence derived from Eq. (6), the microkinetic model predicts a square-root dependence between the induced charge and the light intensity. The latter offers, however, a notably poorer fit to our data (see Supplementary Fig. 2). Additionally, because in the microkinetic model the amount of charge stored on the particle is solely determined by the relative rates of oxidation and reduction, it predicts an infinite accumulation of charges on the particle in the situation characterized by the absence of electron acceptor discussed here. This prediction obviously contradicts the data in Figs. 1 and 4, which clearly show saturation of the charge accumulation. For these reasons, the nanocapacitor picture is the more accurate way to describe the charge accumulation on nanoparticles during redox reactions.

After having deduced an expression for the light intensity dependence of the accumulated charge $\Delta N_{e,\,max}$, we now turn our attention to the charging time constant $\tau$. An obvious approach to

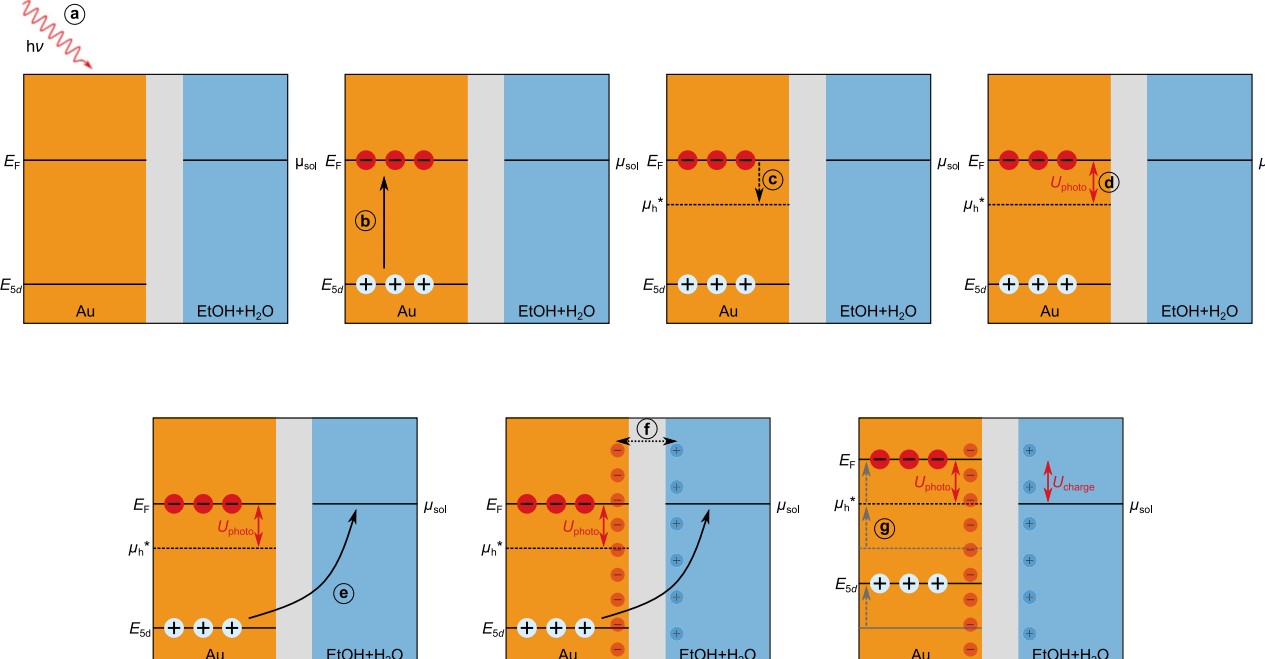

**Fig. 5 | Schematic illustration of the photocharging process.** Absorption of light **a** generates electron--hole pairs on the gold nanoparticles (**b**). Due to the non-thermal excitation of holes in the $5d$-band, the quasi Fermi level of the holes $\mu_h^*$ shifts to lower energies (**c**), inducing a voltage $U_{photo}$ relative to the redox potential of the hole scavenger ethanol (**d**). The latter acts as a driving force for electron transfer from ethanol to the particle (**e**). The charge accumulation is stabilized by the formation of an electric double layer (**f**). Due to the charge accumulation, an additional potential $U_{charge}$ evolves that elevates the energy levels inside the gold (**g**) until eventually, $\mu_h^*$ and $\mu_{sol}$ are in equilibrium.

describe the limited hole transfer rate from gold to EtOH is introducing an activation barrier with a corresponding activation energy $E_A$. This barrier could for example result from a Schottky-like contact formation at the particle surface[49] or from a high-energy reaction intermediate during the EtOH oxidation. In any case, shifting the quasi-Fermi level of the holes modifies $E_A$ for the hole-transfer process by shifting initial potential of the holes in the particle. We therefore write the intensity-dependence of the activation energy as $E_A(I) = E_A' - \mu_h^*(I) = \alpha + k_B T ln(I)$, where $E_A'$ and $\alpha$ denote constants that do not depend on the light intensity. With this expression, the charging time constant is given by:

$$\tau = \frac{1}{k} = \left( k_0 + e^{\frac{E_A}{k_B T}} \right)^{-1} = \left( k_0 + I e^{\frac{\alpha}{k_B T}} \right)^{-1}. \quad (7)$$

Eq. (7) displays an exponential temperature dependence, which potentially has a large influence on the rate. To estimate the influence of possible photoheating, we measured the increase in the solution temperature during the reaction by a thermocouple immersed in the solvent. Even at the highest intensity, the temperature of the solution only increased by 0.9 K during the reaction. Such a low photoheating is not expected to have a significant influence on the reaction rate. Moreover, the intensities used in our experiments are too low to generate local "temperature spikes" around the particles[50,51].

A fit of Eq. (7) to the data is plotted in Fig. 4c (dashed orange line). To obtain a reasonable fit, we had to introduce a dark rate $k_0$. It likely results from the oxidation of EtOH by holes close to the Fermi level, as discussed in the next section. In general, Eq. (7) describes $\tau$ reasonably well, but does not fit as well to the data as the proposed model for $\Delta n_e$. We assume that the reduction of the activation energy due to a light-induced shift of the chemical potential for holes presents only a first-order approximation of the charge transfer process. For a better description, further effects, such as a reduction of the Schottky barrier

by the accumulated charge, a multi-electron transfer process[33], or non-Nernstian electron transfer[52] must be taken into account.

## Influence of EtOH concentration on the charge accumulation

In the following, we want to investigate the influence of the EtOH concentration on the photoinduced charge. To this end, we recorded the resonance evolution for EtOH concentrations from 3.55 M down to 0 M under illumination with 820 mW. The corresponding resonance shifts for the various EtOH concentrations are presented in Fig. 6a. The variations of the shift between the different concentrations are about one order of magnitude smaller than the shift itself. A magnified view along the $y$-axis reveals the slight tendency of an increasing shift with rising EtOH concentration (inset, Fig. 6).

At a first glance, this appears counterintuitive given that the oxidation potential of EtOH is more favourable than that of water[33]. Thus, one might suspect that the voltage between the photoexcited holes and the solvent depends on the EtOH concentration. On the other hand, we discussed in the previous section that the photovoltage $U_{eff}$, and hence $\Delta N_e$, does not depend on the oxidation potential of the solution $\mu_{sol}$. This results from the fact that the electrochemical potential of the particle (in dark) always equilibrates with the environment[40]. The chemical potential of electrons in the solution, $\mu_{sol}$, corresponds to the oxidation potential of the mixture of EtOH and water. The standard oxidation potential of EtOH is $\mu_{EtOH} = -4.66$ eV (0.22 V vs. standard hydrogen electrode), while the oxidation potential of water at pH 7 is $\mu_{H_2O} = -5.26$ eV[53]. Accordingly, the oxidation potential of the water:EtOH mixtures is located between those values, and the tendency to donate electrons to the particle increases with increasing EtOH concentration. However, both potentials are located above the work function of gold at $E_F = -5.3$ to $-5.5$ eV, depending on the crystal facet[53,54]. The difference between the solvent oxidation potential and the particles' Fermi level drives charges into the particles until the Fermi level of the particles is in equilibrium with $\mu_{sol}$[40]. In other words, the particles pick up charge already before illumination[40].

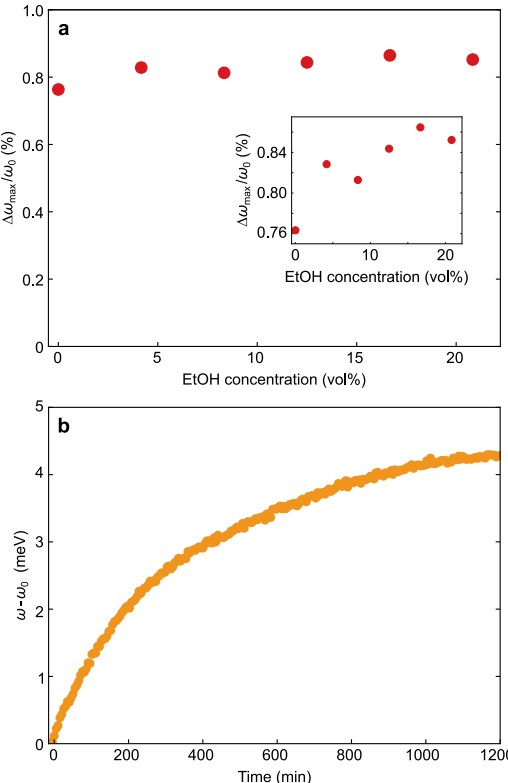

**Fig. 6 | Influence of the EtOH concentration. a** Relative resonance shift $\Delta\omega/\omega_0$ for varying EtOH concentration (red circles). The inset shows a zoom of the $y$-axis to visualize a slight increase in resonance shift for increasing EtOH concentrations. **b** Evolution of resonance positions over 20 h for a sample with 2.84 M EtOH (orange circles). Source data are provided as a Source data file.

In fact, the presence of a permanent charge, and the corresponding electrical double layer, is one of the well-known stabilizing mechanisms for colloidal nanoparticles[45].

Thanks to the sensitivity of the longitudinal resonance to the charge on the nanoparticle, we are able to confirm this equilibration of the chemical potentials. We monitored the evolution of the longitudinal LSPR after adding EtOH to the particles that were otherwise dissolved in water. Again, we removed the oxygen from the solution before recording the spectrum without any illumination. Figure 6b displays the evolution of the longitudinal LSPR over the course of 20 h. The observed blue shift is a direct manifestation of the charge transfer as a result of potential equilibration.

In the context of the photocharging measurements, this implies that after equilibration, $U_{photo}$ does not depend on the EtOH concentration. Consequently, the slightly different values for $\Delta\omega_{max}/\omega_0$ in Fig. 6a indicate that the potential equilibration has not been completed at the start of the illumination. This is consistent with a dark charging time on the order of about 20 h, displayed in Fig. 6b. The higher the illumination power, and hence the photoinduced shift, the less significant this "dark-charging" becomes compared to the light-induced charging effect. As the photoinduced charging was about two orders of magnitude faster than the dark charging, we treated the latter as constant offset during the evaluation of the photoinduced charging. Moreover, the dark charging manifested itself as dark rate $k_0$ in Eq. (7).

## Parameters influencing the charge accumulation

In the following section, we will discuss the influence of several other parameters with practical relevance for photocharging of nanoparticles. These include the particle size, the nature of the ligand, and

the presence of oxygen during the charging process. All measurements shown in Fig. 7 were performed with an excitation at 820 mW illumination power. Figure 7a presents the photoinduced LSPR shift of our standard sample (blue circles) compared to the shift for particles having the same aspect ratio but a 2.5 times larger transverse diameter (red squares) under otherwise identical conditions. With a relative shift of $\Delta\omega_{max}/\omega_0 = 0.24\%$, the larger particles show a smaller total shift than the smaller particles with a shift of $\Delta\omega_{max}/\omega_0 = 0.7\%$. This observation can be rationalized by considering that $\Delta\omega_{max}/\omega_0 \propto \Delta N_{e, max}/N_{e, 0}$. The total charge in the denominator scales with the volume of the particle: $N_{e,0} \propto V$, while the added surface charge in the numerator is proportional to the capacitance Eq. (6), which in turn scales with the surface: $\Delta N_{e, max} \propto A$[49]. Consequently, the relative shift scales with $A/V$. For the present AuNR dimensions, this relationship predicts a 2.5 times higher capacitance for the smaller rods, which is very close to the observed ratio of 2.9. The slight deviation is likely caused by the approximation of the double layer as simple Helmholtz layer. Note that the total number of charges accumulated on the larger particles is larger. However, the relative increase in the number of charges, which determines the magnitude of the LSPR shift, is smaller for the larger particles because of their larger volume.

Next, we investigated the influence of the ligand on the amount of stored charges. Figure 7b contrasts the photo-induced LSPR shift for our standard PVP-coated AuNRs (blue squares) to the shift for AuNRs coated with polyethyleneimine (PEI, orange squares) with the same size and also otherwise identical parameters. We expect $C$, and therefore also $\Delta N_{e, max}$, to be proportional to the dielectric constant $\varepsilon$ of the surface adsorbate layer, which consists mainly of the ligand. PEI possesses a relative permittivity of $\varepsilon_{PEI} \approx 3.3$[55], while PVP's permittivity is about a factor 2.3 higher: $\varepsilon_{PVP} \approx 7.7$[32]. The relative maximum shift of the PVP-coated particles is a factor 2.5 higher than of the PEI-coated AuNRs, which show $\Delta\omega_{max}/\omega_0 = 0.28\%$. This observed value is surprisingly close to the relative difference in permittivity, considering that we did not take into account differences in coating thickness, water content, etc.

As most plasmon assisted chemical experiments in literature are performed under aerobe conditions, we evaluate the influence of oxygen on the accumulated charge. Figure 7c compares the resonance evolution for our standard sample, i.e., under anaerobic conditions (blue circles), to a sample where the oxygen was not removed from the solution (purple squares). As discussed earlier, oxygen works as an efficient electron sink, discharging the particles. Hence, it is not self-evident that, also in the presence of oxygen, a clear, albeit smaller, charging of the AuNRs was observed. The observed shift implies that the photoinduced reduction of the AuNRs by EtOH is faster than their oxidation by molecular oxygen. This is similar to the situation discussed earlier by Kim et al.[22], who used $[Fe(CN)_6]^{3-}$ instead of molecular oxygen as oxidation agent. Within the nanocapacitor model, the oxidation process can be regarded as a leakage current, which removes a part of the stored charge. The equilibrium charge, therefore, not only depends on the particles' capacitance but also on the presence of a charge removal process.

Consequently, the presence of oxygen causes a weaker charging of the rods. This experiment underscores that care must be taken when designing systems for plasmon-induced catalysis. If, for example, a reduction reaction is to be modified by photocharging the metal catalyst, the presence of oxygen manifests another energy decay channel, impeding an efficient reduction.

In conclusion, we investigated the photochemical charging of gold nanorods in situ, by tracking the shift of the longitudinal LSPR during illumination. The presented results clearly demonstrate the accumulation of charges on a metal nanoparticle as a result of photochemical process, which could be rationalized by treating the

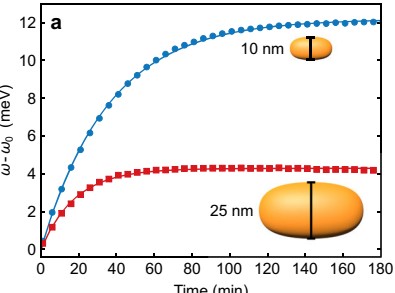
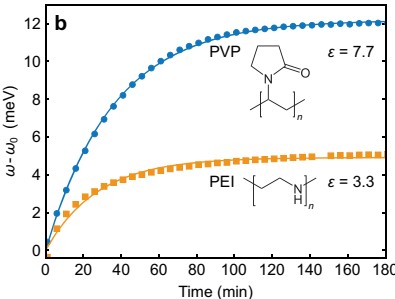
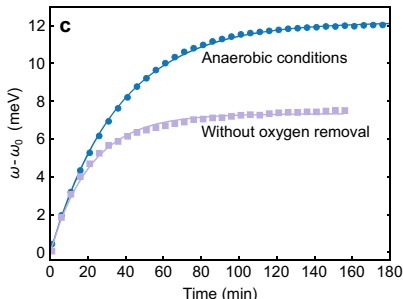

**Fig. 7 | Influence of particle size, ligand and oxygen removal on the photocharging.** In all panels, blue circles represent the reference experimental conditions (anaerobic, 10 nm transverse diameter, and PVP as ligand) under a radiation with 820 mW. In (**a**), the red squares show the LSPR evolution of larger particles (25 nm transverse diameter, same aspect ratio). In (**b**), the orange squares represent the LSPR evolution of particles stabilized with PEI instead of PVP. In (**c**), the light purple squares show the LSPR evolution of a sample without oxygen removal. All measurements were performed under conditions that were otherwise identical to the reference conditions. Source data are provided as a Source data file.

particles as nanoscale capacitors. We introduced a model to assess the number of additional electrons that reside on the particle, which all accumulate on the particle surface. The resulting repulsive potential elevates the whole band structure on the particle, including the Fermi level. This inevitably leads to modifications of the interfacial charge transfer between the particle and its surroundings. The demonstrated photochemical charging, therefore, should be taken into account for envisioned devices relying on plasmon-induced charge transfer, including plasmonic sensors, photovoltaic devices, and, in particular, plasmon-driven photocatalysis.

We captured the light-induced charging potential by introducing a quasi-Fermi level for $5d$-band holes excited via interband transitions. We further corroborated the nanocapacitor picture by investigating the influence of the solvent, the ligand, and the particle size. From this description, several recommendations for achieving high photo-induced charge densities can be derived. In a given reactive environment, one should (i) choose a ligand with high permittivity, thus rather a high $\varepsilon$ polymer than small molecules, (ii) use rather smaller and rather rod-shaped than spherical particles, (iii) use high laser powers, and (iv) prevent the presence of unwanted electron scavengers, in particular oxygen.

## Methods

### Materials
All nanorods were purchased as colloidal solutions from Nanopartz. Lot numbers are provided in the section "sample preparation". DI water was used in all experiments and was further purified using an ELGA (purelab classic) ultrapure water system (Milipore, 20 MΩ•cm). EtOH was purchased from Carl Roth (99.8% p.a., anhydrous, ROTIPURAN). Thionin-acetate was purchased from Thermo Scientific in powder form (Lot# 10236119, dye content 90%). EtOH and thionine acetate were used without further purification. The thionin acetate powder was dissolved in DI water by vortex shaking to obtain a 55 mM solution.

### Characterization
Quartz cuvettes containing a gold nanorod (AuNR) dispersion were placed inside a modified UV/VIS spectrometer (Cary 5e) that allowed the illumination of the sample with a laser, while absorbance spectra were simultaneously recorded perpendicular to the excitation beam.

### Sample preparation
We employed AuNRs with a transverse diameter of 10 nm and a nominal longitudinal plasmon resonance at 700 nm. The particles, stabilized by a layer of PVP ($M_w$ 40 kg mol⁻¹). Note that particles from three different fabrication batches were used during the experiments (batch 1: Lot# L9074, batch 2: Lot# N1442, batch 3: Lot# P4464). The batches apparently possessed slight differences in shape, causing slight differences in

longitudinal LSPRs (batch 1: 706 nm, batch 2: 676 nm, batch 3: 700 nm). In particular, the particles used for Fig. 1d,e were taken from batch 2, the particles used for Fig. 3 were taken from batch 3. In the other cases, batch 1 was used. For each sample, 150 μL of particle solution (as purchased) was added to a mixture of DI water and EtOH, eventually resulting in 2 mL solution. If not stated differently, the total EtOH concentration was 2.84 M. Only for the measurements presented in Fig. 3, 300 μL AuNR dispersion was used. The samples were prepared in quartz cuvettes (Hellma, cubic base with an edge length of 1 cm) that had previously been purged with nitrogen. To remove oxygen from the solution and ensure anaerobic conditions, the samples were bubbled with nitrogen for 45 min prior to each measurement. The bubbling also ensured a thorough mixing of the solution.

For the investigation of the influence of the ligand (Fig. 7b), we purchased PEI-stabilized nanorods ($M_w$ 100 kg mol⁻¹, Lot# N1443) with a transverse diameter of 10 nm and a nominal longitudinal LSPR at 700 nm. For the investigation of the influence of the size (Fig. 7c), we purchased nanorods with a transverse diameter of 25 nm and a nominal longitudinal plasmon resonance at 700 nm (Lot# 1444). In both cases, 150 μL of particle solution (as purchased) was mixed with 1518 μL DI water and 332 μL EtOH.

### Photocharging measurements
The solutions were excited by a 532 nm laser (Laser Quantum finesse 532 pure) with various laser powers. The laser cross-section at the cuvette was 1 cm². The absorbance spectrum was recorded every 5 min during excitation perpendicular to the laser beam. To ensure homogeneous measurement conditions, the solution was stirred during all measurements. The temperature of the solution was monitored during the reaction using a PT1000 thermocouple, located in the solution but outside the illumination spot. The scattered laser radiation was blocked by a notch filter. This allowed us to record the full longitudinal LSPR of the AuNRs ≈700 nm, but masked the transverse LSPR at ≈550 nm.

### Capacitance calculation
To obtain the nanorod capacitance, we used the proportionality $n_h^* = \zeta I$ (with $\zeta$ being a proportionality constant) to rewrite Eq. (6) to the form:

$$\frac{\Delta N_{e,\,max}}{N_{e,\,0}} = \alpha + \beta \ln(I). \tag{8}$$

Consequently, we find an expression for the relative resonance shift as

$$\frac{\Delta \omega_{max}}{\omega_0} = a + b \ln(I) \tag{9}$$

with the fitting constants

$$a = 0.7 \frac{C}{q_e^2 V n_{e,0}} \cdot \left[ \mu_{EtOH} - E_{5d} + k_B T \cdot ln\left(\frac{\zeta}{N_{5d}}\right) \right] \quad (10)$$

and

$$b = 0.7 \frac{C k_B T}{q_e^2 V n_{e,0}} . \quad (11)$$

From the fitting constant $b$ (which is 0.0025 in our fit), we are able to deduce the capacity $C$ of the particles, as it otherwise only depends on the easily accessible parameters temperature $T$, particle volume $V$ and initial electron density $n_0$ and natural constants. We assume the particles to be cylinders with hemispheres on both ends with a radius of 5 nm and a total length of 29 nm (as specified by the manufacturer for batch 1). With an initial electron density[56] of $n_{e,0} = 5.9 \cdot 10^{22}$ cm$^{-3}$ and a temperature of 293 K, we obtain a capacity of $2.7 \cdot 10^{-15}$ F. This translates to a value of 294 µF cm$^{-2}$.

## Data availability
The data that support the findings of this study are available from Zenodo[57] and from the corresponding author upon request. Source data are provided with this paper.

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

## Acknowledgements
F.S., M.B., and W.K. acknowledge funding by the Deutsche Forschungsgemeinschaft (DFG, German Research Foundation) - CRC/SFB 1636 - Project ID 510943930, Project No. A01 / A04.

## Author contributions
F.S. and W.K. drafted the project, conducted the experiments, evaluated the data, and co-wrote the manuscript. M.B. supervised the work and worked on the manuscript.

## Funding

## Competing interests
The authors declare no competing interests.
