## [Transparent Peer Review file · Nature Communications]

Capacitive Photocharging of Gold Nanorods

Corresponding Author: Dr Wouter Koopman

Version 0:

Reviewer comments:

Reviewer #1

(Remarks to the Author)

In this communication, Koopman and coworkers describe the measurement of the accumulated (excess) electron density on plasmonic nanoparticles under photocatalytic conditions. Previously, these densities were determined indirectly through Fermi level shifts or onset potentials. The direct measurements performed here by in-situ LSPR shift measurements are valuable because they allow quantitative confirmation of previously reported trends and model in literature on plasmonic photochemistry and catalysis. However, the manuscript also raises the following doubts and questions, especially about the similarity and differences between the current findings and prior data/model.

1a) The manuscript states "Previous studies that inferred the charging indirectly, agreed on a steady rise of the accumulated charge with increasing illumination intensity. However, no quantitative understanding of the charging mechanism was provided." The latter statement contradicts the prior work (e.g., ref. 22) where a microkinetic model based on asymmetric charge transfer kinetics was developed and used to explain the dependence of the excess charge on the illumination intensity.

1b) The manuscript states "A previous study assumed a square root dependence of the charge density on the illumination intensity". This was not an assumption but an outcome of a microkinetic model based on asymmetric charge transfer. I recommend that in place of these statements, the authors describe the prior model (asymmetric charge transfer kinetics) and compare it with their model.

1c) The observation "This implies that the photoinduced reduction of the AuNRs by EtOH is much faster than the oxidation by molecular oxygen." aligns well with prior observations, e.g., the concentration of the alcohol hole acceptor directly influences the electron density accumulated on photoexcited gold nanoparticles. In fact, the previously reported model (ref. 22) is founded on asymmetry in the kinetics of hole and electron transfer, i.e., electron accumulation is the result of holes being consumed by the alcohol hole acceptor at a rate faster than electrons being consumed by the electron scavenger.

2) It would also be useful to show fits of the experimental data in Fig. 2b to prior model/s so that the readers have context for the statement "This alternative model does not fit to our data." Otherwise, the trend/shape of the plot in Fig. 2b appears similar to the previously reported dependence of the excess electron density on the intensity.

3) How does the excess electron density measured here for Au nanorods compare with that on Au nanospheres at the same illumination power? In other words, to what extent do nanoparticle shape and/or the nature/density ligands influence the degree of charging.

4) The manuscript states "The potential difference between μh^* and the oxidation potential of EtOH, μ_{EtOH} , corresponds to the photoinduced voltage U_{photo} ". I wonder whether it is correct to term this as the photoinduced voltage. In principle, in the dark, the photoinduced voltage should be 0. Is that the case for the quantity in question here? In past work (e.g., ref. 16), the photoinduced potential is defined by a difference between the chemical potential under light and that under dark.

5) The manuscript states "Naively, one could therefore expect μh^* to be equal to the upper edge of the 5d-bands." I wonder why this would be expected. Rather, one would expect μh^* to be dependent on the steady-state concentration of holes, which in turn would be dependent on the illumination intensity, the hole acceptor concentration, and the oxidation potential

of the hole acceptor.

6) In past work, the hole acceptor concentration has been shown to affect the excess electron density. Is this also the author's finding?

Reviewer #2

(Remarks to the Author)

In this manuscript, the authors report an experimental study in which charging of plasmonic nanoparticles, upon resonant illumination and in presence of a hole scavenger, is studied using the induced changes in optical properties. The article is well written and logically constructed, the figures are of high clarity and quality, the use of references is appropriate, the authors are well aware of the state-of-the-art, and the experimental results are of high quality and with important practical impact in the field of plasmonics. Also, the data on photocharging under oxygenated vs deoxygenated conditions, for different nanoparticle sizes, and in presence of different capping agents is valuable to the community. Finally, the recommendations at the end of the article are very sound and helpful.

However, the phenomenon of nanoparticle charging and its influence on plasmonic properties and photocatalysis is already very well documented since a long time, for instance by Mulvaney and coworkers in 2006 (already cited by the authors) and extensively reviewed by for instance Scanlon & Girault et al. (10.1039/C5SC00461F). Further, photocharging and discharging of plasmonic systems upon redox chemical activity, and monitoring this using absorption measurements, was also already reported by Jain et al. in 2013 (10.1021/jz401719u) and is also present in the data of Jain et al. from 2018 (10.1038/s41557-018-0054-3).

Therefore, although it must be emphasized that the scientific quality is high, it is this reviewer's opinion that the findings and methodology presented here are not sufficiently novel enough to warrant publication in Nature Communications.

Reviewer #3

(Remarks to the Author)

See review attached.

Reviewer #4

(Remarks to the Author)

The authors discuss the in situ observation of nanoparticle photocharging, specifically focusing on gold nanorods. The study introduces a method to track the charging of gold nanorods during a light-induced reaction by monitoring the longitudinal plasmon resonance of the rods. The results provide spectroscopic evidence that the charging process can be understood as a nanoscale capacitor model. The authors also analyze the dependence of the charging process on particle size, oxygen content in the solvent, and ligand type.

The manuscript is well-written and the topic is of high interest. The findings regarding the rational engineering of dynamic charge states can be relevant for plasmon-driven photoreactions.

The model the authors propose is simple yet apparently quite effective. The authors also considered potential alternatives to the observed shift and described the reasons behind the identification of photocharging as the main mechanism at play.

I have some comments that should be addressed before recommending the publication:

- At lines 186-189 the authors mention, to prove the origin of the resonant shift, that after centrifuging, the same red-shift was observed. Do they mean blue-shift? Or maybe they meant the same red-shift vs. time from the previously blue-shifted resonance? In any case, did the authors verify that, upon switching the laser off and resetting the environment, the same resonance (blue) shift was present and that shift eventually progressed toward the original resonant wavelength?
- Did the authors try the same intensity-dependent experiments with different ethanol concentrations? This should be done to (i) provide a baseline for comparing potential shifts due to other effects (corroborating the authors' discussion) and (ii) analyze the impact on charging dynamics. Is there a concentration for which ethanol concentration becomes the bottleneck of the process compared to the rate at which electrons are generated? The authors should discuss the concentration effects further.
- For completeness, the authors should provide an estimate of the temperature increase of the solution (depending on illumination power) following, for example, 'Experimental and Theoretical Studies of Light-to-Heat Conversion and Collective Heating Effects in Metal Nanoparticle Solutions': (<https://pubs.acs.org/doi/epdf/10.1021/nl8036905>) and discuss its potential impact (or the lack of) on the experiments.
- There is a typo 'Remval' in Fig. 4c

Version 1:

Reviewer comments:

Reviewer #1

(Remarks to the Author)

Most of the questions or suggestions in my report have either been addressed or a valid explanation has been provided for

the lack of clear resolution at this time.

There is one remaining aspect that deserves revision for the purpose of accuracy. In prior work, as the authors point out, Coulomb repulsion was not explicitly included in the microkinetic rate expressions; however, there was a limit to charging due to two factors that were qualitatively discussed in the model:

- (i) The limit due to the double-layer capacitance of the metal nanoparticle, which is a function of the dielectric permittivity of the medium (besides other factors), was discussed in ref. 22. To directly quote an excerpt "Such large electron density changes can be accommodated due to the high double-layer capacitance of Au nanoparticles in water, the magnitude of which is two orders greater than in vacuum. Possibly, the magnitude of the double layer capacitance is the factor limiting the maximum cathodic polarization achievable in the event the hole scavenging and photoexcitation conditions are ideal."
 - (ii) In the absence of an electron acceptor, the accumulated electrons will slowly react with protons at the interface, which are formed by the hole-driven oxidation of the alcohol. This phenomenon was explicitly demonstrated in ref. 27 in the form of H₂ evolution as a alternative pathway to accumulated electron capture by the electron acceptor. This was further leveraged in <https://pubs.acs.org/doi/10.1021/acseenergylett.9b01688> where it was shown that pH of the medium controls the relative contribution of this pathway. This side reaction is yet another factor that limits electron buildup.
- Given the recognition of these factors, it is incorrect to state that prior models allow for infinite charge accumulation.

Reviewer #3

(Remarks to the Author)

I am fully satisfied with the answers provided by the Authors to my and other reviewer's reports. The paper can be published now.

Reviewer #4

(Remarks to the Author)

The authors have replied to my comments and also added material, in particular to the discussion related to the EtOH concentration dependence. Provided that the comments of the other reviewers have also been addressed satisfactorily, I can recommend the publication of the manuscript.

REVIEWER REPORT(S):

Reviewer # 1

In this communication, Koopman and coworkers describe the measurement of the accumulated (excess) electron density on plasmonic nanoparticles under photocatalytic conditions. Previously, these densities were determined indirectly through Fermi level shifts or onset potentials. **The direct measurements performed here by in-situ LSPR shift measurements are valuable because they allow quantitative confirmation of previously reported trends and model in literature on plasmonic photochemistry and catalysis.** However, the manuscript also raises the following doubts and questions, especially about the similarity and differences between the current findings and prior data/model.

We thank the reviewer for the thorough review that helped us to improve our arguments. We hope that the revised manuscript will convince the reviewer that the current work does not fundamentally contradict previous findings but rather improves upon the earlier work available in the literature and provides important extended insights that will help to further enhance plasmon catalysis.

1a) The manuscript states “Previous studies that inferred the charging indirectly, agreed on a steady rise of the accumulated charge with increasing illumination intensity. However, no quantitative understanding of the charging mechanism was provided.” The latter statement contradicts the prior work (e.g., ref. 22) where a microkinetic model based on asymmetric charge transfer kinetics was developed and used to explain the dependence of the excess charge on the illumination intensity.

We appreciate the reviewer pointing out that this sentence is misleading. Indeed, the Jain group (e.g. ref 22) provided a microkinetic model that calculates a number of charges which accumulate on a particle due to an imbalance in the oxidation and reduction rates. This kinetic model intrinsically treats electrons as neutral as it does not include the influence of the electric field of the electrons accumulated on the particle surface on the potential (e.g. the repulsive forces) seen by the electrons. In this sense, the earlier model does not consider the charging, but only the accumulation of charge carriers. In our opinion this does not adequately reflect reality in some important aspects (see also answer to 1c). The criticized sentence was meant to reflect this.

In the revised version we completely rewrote the introduction to make the difference of our work to earlier works clearer.

1b) The manuscript states “A previous study assumed a square root dependence of the charge density on the illumination intensity”. This was not an assumption but an outcome of a microkinetic model based on asymmetric charge transfer. I recommend that in place of these statements, the authors describe the prior model (asymmetric charge transfer kinetics) and compare it with their model.

We thank the reviewer for the suggestion and reworked the section accordingly to lay out the differences between our model and the previous microkinetic model. We included the comparison in the “Intensity dependence of charging: capacitor model” section on pages 6-7.

1c) The observation “This implies that the photoinduced reduction of the AuNRs by EtOH is much faster than the oxidation by molecular oxygen.” aligns well with prior observations, e.g., the

concentration of the alcohol hole acceptor directly influences the electron density accumulated on photoexcited gold nanoparticles. In fact, the previously reported model (ref. 22) is founded on asymmetry in the kinetics of hole and electron transfer, i.e., electron accumulation is the result of holes being consumed by the alcohol hole acceptor at a rate faster than electrons being consumed by the electron scavenger.

In the revised version we include a detailed discussion of the previous microkinetic model in comparison to our nano-capacitor model (see also answer 1b). We do not believe that the microkinetic model is fundamentally wrong. However, its neglect of the Coulomb repulsion between electrons results in too large predictions for the number of accumulated charges. In the microkinetic model, charge accumulation is fully determined by the relative rates of oxidation and reduction. As a result, it predicts an infinite accumulation of charge carriers on the particle when applied to the situation of a vanishing oxidation rate which was studied in our manuscript. This contradicts both intuition and our data, from which we conclude that the previous model is incomplete.

2) It would also be useful to show fits of the experimental data in Fig. 2b to prior model/s so that the readers have context for the statement “This alternative model does not fit to our data.” Otherwise, the trend/shape of the plot in Fig. 2b appears similar to the previously reported dependence of the excess electron density on the intensity.

As requested, we now include a comparison of a best fit for the square root dependence of the microkinetic model and the logarithmic dependence of the nanocapacitor model in the SI of our manuscript (see Supplementary Note 3).

3) How does the excess electron density measured here for Au nanorods compare with that on Au nanospheres at the same illumination power? In other words, to what extent do nanoparticle shape and/or the nature/density ligands influence the degree of charging.

The reviewer raises an important point. Indeed, a certain dependence of the capacitance on the geometry of the particle can be expected. Unfortunately, the available setup does not allow us to measure the charging of a gold nanosphere. Therefore, the only experimental data we can provide is the comparison of the resonance shift for rods of two different sizes, which are consistent with the expected variation of the particles' capacitance (see figure 6a in the revised manuscript).

In the revised version, we include an estimation of the relative increase of the areal capacitance of a nanorod compared to capacitance of nanospheres published by the Tschulink group (10.1002/anie.202112679). The value we obtain for the nanorod (290 $\mu\text{m}^2/\text{cm}^2$) is about three times larger than the capacitance published for a sphere ($\approx 100 \mu\text{m}^2/\text{cm}^2$) which is consistent with the expected increase of the areal capacitance for an ellipsoidal capacitor compared to a spherical capacitor (see. Sandu 2013, 10.1063/1.4847495 and Momoh 2009, 10.1002/mop.24630). We included this estimation in the section “Intensity dependence of charging: capacitor model” on page 6.

Regarding the influence of the ligands, Figure 6b of our manuscript compares the charging with two different types of ligands (PVP vs PEI). We interpret the observed difference as a confirmation that the capacitance of the particles is strongly determined by the dc permittivity of the ligand shell. In future, a systematic study on the ligands' influence should be conducted. This is however out of the scope of the current manuscript.

4) The manuscript states “The potential difference between μh^* and the oxidation potential of EtOH, μEtOH , corresponds to the photoinduced voltage U_{photo} ”. I wonder whether it is correct to term this as the photoinduced voltage. In principle, in the dark, the photoinduced voltage should be 0. Is that

the case for the quantity in question here? In past work (e.g., ref. 16), the photoinduced potential is defined by a difference between the chemical potential under light and that under dark.

The definition of the U_{photo} is indeed an important point in our manuscript. The thermodynamic potential of the excited holes is given by the quasi Fermi level μ^* of excited holes. In the dark, this quantity is equal to the Fermi level of the electrons (E_F). On the other hand, the driving force for the light-driven charge transfer between particle and solution is given by the difference between μ^* and the oxidation potential of the EtOH / Acetaldehyde redox couple (i.e. μ_{sol}). In the previous manuscript, we identified this driving force as U_{photo} . In a thermodynamic equilibrium (in the dark) the Fermi level of the particle is identical to μ_{sol} . In our case that is equal to μ_{EtOH} . In the dark the driving force thus vanished indeed, as expected for U_{photo} .

After careful consideration of the reviewers' reasoning, we agree that this designation is misleading and the photovoltage is different from the driving force for charge transfer (i.e. $\mu_{\text{sol}} - \mu^*$). The latter decreases due to the charging process, while U_{photo} should exclusively depend on the absorbed light. We therefore revised the definition of the photo-voltage to correspond to the more general definition $U_{\text{photo}} = (E_F - \mu^*)/q_e$. With this definition, U_{photo} remains constant, because the potential induced by the charge on the surface lifts the entire band-structure (including E_F and μ^*) to higher energies relative to μ_{sol} , until $\mu^* = \mu_{\text{sol}}$. In the revised manuscript, we extended Figure 4 as visual aid to this explanation.

5) The manuscript states "Naively, one could therefore expect μ^* to be equal to the upper edge of the 5d-bands." I wonder why this would be expected. Rather, one would expect μ^* to be dependent on the steady-state concentration of holes, which in turn would be dependent on the illumination intensity, the hole acceptor concentration, and the oxidation potential of the hole acceptor.

We agree with the reviewer that it is probably better not to make a statement about what "one" may expect, since this strongly depends on the reader. We therefore deleted the sentence, as it is not relevant for our message.

Indeed, μ^* is a measure for the steady state concentration of holes, as we state in the manuscript when introducing the concept of quasi Fermi-levels. However, we often come across the (sometimes implicit) expectation that the relevant potential difference for driving holes from the particle to a solution (or molecule) is the difference between the energy of the 5d-band and the oxidation potential of the solution. (Of course, if that was true, the saturation charge should be independent of the illumination intensity – which is clearly not the case.) The sentence was meant to pick up readers with this idea.

6) In past work, the hole acceptor concentration has been shown to affect the excess electron density. Is this also the author's finding?

The reviewer raises an interesting point. In the revised manuscript we added a series of measurements for different hole scavenger concentration. The measurements indicate no difference for the saturation resonance shift $\Delta\omega/\omega_0$ for different EtOH concentrations. This behavior can be understood when considering that a metallic NP immersed in a solution will reach an electrochemical equilibrium with the surrounding solution. This implies that the Fermi-level of the particle equilibrates with the solvent, in this case the oxidation potential of the EtOH / Acetaldehyde redox couple, μ_{EtOH} (see also the review of Scanlon & Girault, *Chem. Sci.* **2015**, 6 (5), 2705–2720. <https://doi.org/10.1039/C5SC00461F>).

In our opinion, this difference to earlier measurements can be understood from the fact that we investigated the maximum storable charge, while in earlier works by Jain and others, the charging was strongly influenced by the relative rates of oxidation and reduction. A change in the hole acceptor concentration changes the related chemical potential and as a result also the rate of hole

transfer. Through this mechanism a change in the hole acceptor concentration can be expected to influence the effective charge accumulation. In our investigation, the rate of electron transfer from the particle is negligible and hence does not influence the accumulation of electrons on the particle.

Reviewer # 2

In this manuscript, the authors report an experimental study in which charging of plasmonic nanoparticles, upon resonant illumination and in presence of a hole scavenger, is studied using the induced changes in optical properties. The article is well written and logically constructed, the figures are of high clarity and quality, the use of references is appropriate, the authors are well aware of the state-of-the-art, and **the experimental results are of high quality and with important practical impact in the field of plasmonics**. Also, the data on photocharging under oxygenated vs deoxygenated conditions, for different nanoparticle sizes, and in presence of different capping agents is valuable to the community. **Finally, the recommendations at the end of the article are very sound and helpful.**

However, the phenomenon of nanoparticle charging and its influence on plasmonic properties and photocatalysis is already very well documented since a long time, for instance by Mulvaney and coworkers in 2006 (already cited by the authors) and extensively reviewed by for instance Scanlon & Girault et al. (10.1039/C5SC00461F). Further, photocharging and discharging of plasmonic systems upon redox chemical activity, and monitoring this using absorption measurements, was also already reported by Jain et al. in 2013 (10.1021/jz401719u) and is also present in the data of Jain et al. from 2018 (10.1038/s41557-018-0054-3).

Therefore, although it must be emphasized that the scientific quality is high, it is this reviewer's opinion that the findings and methodology presented here are not sufficiently novel enough to warrant publication in Nature Communications.

We want to thank the reviewer recognizing the high scientific quality of our work. We hope that we can convince the reviewer that indeed novel aspects of our work should be valued higher.

We are aware that a lot of work has been conducted on both the charging of nanoparticles and also the methodology to measure this charge by shifts in the surface plasmon resonance. However, as also reviewer #3 explains in much detail, these previous works have fundamental flaws in the sense that they do not take into account the specific behavior of charges on a metal sphere. That is, due to the free movement of charges in a metal, the repulsive forces between the charges will accumulate them on the surface of the particle. In effect, the charge in the bulk of the particle is not modulated by adding charges to a metal particle. This fact is neither recognized in the work of Mulvaney (and in the review by Scanlon & Girault who cite this work) nor in the work of Jain. On the contrary, the work of Mulvaney explicitly assumes that the bulk density of charges in the particle is modulated, which in turn modifies the plasmon resonance via the Drude part of the dielectric function. This work has therefore been strongly criticized, see e.g. Langmuir 32, 2829–2840 (2016), and the impact of a surface charge on the plasmon resonance is an active field of research (e.g. Zurak, et al *Science Advances* 2024, 10 (36), eadn5227).

Also, the work by Jain et al. on the impact of charging on photocatalyzed redox chemistry falls short of recognizing the importance of Coulomb forces. Therefore, it overestimates the number of charges that are accumulated on the particle. Moreover, while the approach to calculate the charging via the imbalance of oxidation and reduction rates makes an important point, it does not consider the

repulsive force of the accumulated charge, counteracting the charging process. Finally, the idea presented in this work, that the Fermi level of the metal particles changes by “filling” gold particles with electrons, implies that the Fermi level would have to shift in relation to the 5d-bands of gold. As a consequence, the optical absorption by interband transitions would significantly change – which has not been observed.

Therefore, it can be stated that the current state of the literature does not represent reality in a very fundamental point, which may hinder the systematic exploitation of dynamic Fermi level engineering in photocatalysis with plasmonic particles.

Our nano-capacitor approach naturally recognizes both, i) the accumulation of particles at the surface and ii) the influence of the Coulomb forces on the charge accumulation. Also, the electrochemical potential of the particles is mainly modulated by the Coulomb potential of the charges induced inside the particle. This implies that the entire bandstructure, instead of only the Fermi level, is shifted up or down. A very important implication of the improved understanding is that it can be used to make appropriate recommendations for the design of particles with tailored redox properties. For example, in our manuscript we show a significant impact of the choice of ligand shell (which has not been discussed in previous models) on charge accumulation and potential shift. We are therefore convinced that our work is novel and impactful enough to warrant a publication in Nature Communications.

We agree however that the points made in this answer were not clear in the previous manuscript and therefore revised in particular the introduction to clarify the difference to existing works.

Reviewer #3

The Manuscript reports on the shift of the longitudinal plasmon of gold nanorod upon the photochemical reaction. Thus, the light induced reaction can be followed in real time. The authors attribute the shift of the plasmon resonance energy to the photoinduced charging of the gold nanorod, so that this is the charging process that they pretend to observe in the real time. For the reason given below I recommend rejection of the present work.

Main criticism.

Early works of Mulvaney (see e.g. ref 4, 6, 7 of the present manuscript) lead to an erroneous idea that the charging of metallic nanoparticles would lead to the shift of the plasmon resonance. In turn, the shift of the plasmon resonance can be used to trace the charging. Indeed, the proposal seem simple and attractive. E.g. for the small free-electron metal spherical nanoparticle, the nonretarded dipolar plasmon resonance frequency of the spherical nanoparticle is given by $\omega_{DP} = 1/\sqrt{3} \omega_B$. Here $\omega_b = \sqrt{n}$ is the bulk plasmon frequency (n is the **bulk** electron density, and atomic units are used). Thus, from the change of ω_{DP} one deduces the change in the bulk electron density Δn , and therefore the charge of the nanoparticle $Q = V \Delta n$ (V stands for the nanoparticle volume).

So far so good as far as formal Math is involved. The problem actually comes from the Physics. Classical electrostatics or quantum theory of metals tell us the same: there cannot be any excess charge in the volume of metal nanoparticle. Indeed, by virtue of the Gauss theorem, the presence of the charge in the volume will produce an electric field which would move the electrons until this field is screened. In other words, for the finite size object the charge is rejected from the volume and resides at the surface. This is the so-called Faraday's Ice Pail Theorem.

As follows from the quantum time-dependent density-functional theory calculations performed in Langmuir 32, 2829–2840 (2016) (<https://pubs.acs.org/doi/10.1021/acs.langmuir.6b00112>), the surface charging is not sufficient to induce any appreciable shift of the plasmon resonance, and in any case, this shift does not follow the simple trends predicted by the \sqrt{n} dependence. Therefore, the discussion presented in the manuscript is wrong, and the shift of the plasmon resonance is by no means the measure of the change of the 3D electron density. Most probably it reflects the changes in the medium surrounding the nanoparticle.

We want to thank the reviewer for concisely summarizing the flaws in the current literature on charge induced shift of plasmon resonances. We agree with this analysis of the existing literature but feel that the reviewer did not appreciate the difference to our proposed model: We assume the particles to be nanocapacitors. In a capacitor the charge is naturally located at the surface - not in the bulk. Therefore, the capacitance is not only a function of the particle itself, but also of its surroundings – in particular of the electrical double layer at its surface, that mainly consists of the ligand shell. The work cited by the reviewer, on the other hand, assumes a charged sphere in vacuum, which only has the considerably lower capacitance of the metal itself. In our manuscript, we show that the ligand indeed has a significant influence on capacitance, and hence the charging of the particle. Moreover, landing experiments (Tschulink, 10.1002/anie.202112679) and redox titration experiments (Lyu,) already have proven that in realistic environments, metal nanoparticles can host significantly more charge than if they were in vacuum, the situation assumed in the paper cited by the reviewer.

With respect to the shift of the plasmon resonance, we would like to point out that also the previous version of our manuscript does not use the bulk electron density argumentation by Mulvaney to explain the observed shift in the plasmon resonance. Instead, in the original manuscript we referred to the work of Hoener et al. The authors introduce a core-shell model, in which the charge resides in a multi-layered shell (Hoener, 10.1021/acs.jpcclett.7b00945). To consider all possible influences on the charge accumulation, in the revised manuscript we experimentally measure the charge accumulated by the photoinduced charging process using the electron scavenger thionine. By correlating the decrease in thionine concentration with the shift in LSPR, we can determine the exact relationship between plasmon shift and charge carrier concentration (Figure 3 of the revised manuscript).

We hope that reasoning presented here and the revisions in our manuscript convince the reviewer of the correctness and importance of our work.

Reviewer #4

The authors discuss the in situ observation of nanoparticle photocharging, specifically focusing on gold nanorods. The study introduces a method to track the charging of gold nanorods during a light-induced reaction by monitoring the longitudinal plasmon resonance of the rods. The results provide spectroscopic evidence that the charging process can be understood as a nanoscale capacitor model. The authors also analyze the dependence of the charging process on particle size, oxygen content in the solvent, and ligand type.

The manuscript is well-written and the topic is of high interest. The findings regarding the rational engineering of dynamic charge states can be relevant for plasmon-driven photoreactions.

The model the authors propose is simple yet apparently quite effective. The authors also considered potential alternatives to the observed shift and described the reasons behind the identification of photocharging as the main mechanism at play.

We thank the reviewer for the comments and are glad to hear the relevance of our work is appreciated.

I have some comments that should be addressed before recommending the publication:

- At lines 186-189 the authors mention, to prove the origin of the resonant shift, that after centrifuging, the same red-shift was observed. Do they mean blue-shift? Or maybe they meant the same red-shift vs. time from the previously blue-shifted resonance? In any case, did the authors verify that, upon switching the laser off and resetting the environment, the same resonance (blue shift) was present and that shift eventually progressed toward the original resonant wavelength?

Indeed, we meant to imply that after resetting the chemical environment, the LSPR retained its blue shifted value that it *acquired* during photocharging, followed by a backshift to the *original resonance position before illumination*. We agree that the section was somewhat unclear. In the revised version it reads on p3:

Chemical changes such as a modification of the AuNR geometry or of the environment's permittivity are expected to cause a permanent displacement of the resonance and are therefore probably not the origin of the observed shift. However, the permittivity in the vicinity of the particle might also be modified (electro)chemically. The backshift could then signify the diffusion of the reaction product away from the particle. Also, gold ions that were dissolved under radiation might readsorb as soon as the laser is turned off.²⁴ In order to exclude these processes, we centrifuged the particles immediately after illumination and replaced the supernatant solution with water. After washing, the particles initially retained the blue-shift they had acquired during illumination and subsequently showed the same backshift towards the original resonance position as the unwashed particles (see Supplementary Note 1).

- Did the authors try the same intensity-dependent experiments with different ethanol concentrations? This should be done to (i) provide a baseline for comparing potential shifts due to other effects (corroborating the authors' discussion) and (ii) analyze the impact on charging dynamics. Is there a concentration for which ethanol concentration becomes the bottleneck of the process compared to the rate at which electrons are generated? The authors should discuss the concentration effects further.

The reviewer raises an interesting point. The EtOH concentration should influence the redox potential of the solution and therefore could be expected to have an influence on the driving force for the electron transfer. In the revised version, we added a section in which we discuss the impact of the EtOH concentration on the saturation charge. Interestingly, it turns out that the EtOH concentration does not have an effect on the amount of charge that can be stored on the particle. Our interpretation is, that the particles reach an electrochemical equilibrium with the solution before the start of the measurement (see also the review of Scanlon & Girault, *Chem. Sci.* **2015**, *6* (5), 2705–2720. <https://doi.org/10.1039/C5SC00461F>). In this way, the 5d-band, and hence the magnitude of U_{photo} , which determines the saturation charge, have the same difference to the redox potential of the solution, independent of the actual value of the latter (i.e. of the EtOH concentration).

- For completeness, the authors should provide an estimate of the temperature increase of the solution (depending on illumination power) following, for example, ‘Experimental and Theoretical Studies of Light-to-Heat Conversion and Collective Heating Effects in Metal Nanoparticle Solutions’: (<https://pubs.acs.org/doi/epdf/10.1021/nl8036905>) and discuss its potential impact (or the lack of) on the experiments.

We agree that temperature is generally an important point when dealing with reactions photo-catalyzed by plasmonic particles. In particular, we expected it to have an influence on the charging rate (see eq. 8 in the revised manuscript). To estimate the influence of photoheating, we therefore continuously monitored the temperature during illumination using a thermocouple. The couple was located in solution, but outside the illumination spot. Even for the highest laser power used, the temperature rose only by 0.9 K. We do not expect that such a marginal increase in temperature has a significant influence on the reaction rate. Moreover, due to the low temperature increase “temperature spikes” as discussed e.g. in the articles Baffou, (ACS Nano, 7,8,6478,2013) are very unlikely. We therefore consider photoheating to be not of importance. In the revised manuscript we discuss the photoheating with regard to its influence on the reaction rate. The section on p7 reads:

Equation 7 displays an exponential temperature dependence, which potentially has a large influence on the rate. To estimate it the influence of possible photoheating, we measured the increase of the solution temperature during the reaction by a thermocouple immersed in the solvent. Even at the highest intensity, the temperature of the solution only increased by 0.9 K during the reaction. Such a low photoheating is not expected to have a significant influence on the reaction rate. Moreover, the intensities used in our experiments are too low to generate local ‘temperature spikes’ around the particles.^{55,46}

- There is a typo ‘Remval’ in Fig. 4c

We thank the reviewer for pointing this out. We corrected this typo.

Reviewer #1 (Remarks to the Author):

Most of the questions or suggestions in my report have either been addressed or a valid explanation has been provided for the lack of clear resolution at this time.

There is one remaining aspect that deserves revision for the purpose of accuracy. In prior work, as the authors point out, Coulomb repulsion was not explicitly included in the microkinetic rate expressions; however, there was a limit to charging due to two factors that were qualitatively discussed in the model:

(i) The limit due to the double-layer capacitance of the metal nanoparticle, which is a function of the dielectric permittivity of the medium (besides other factors), was discussed in ref. 22. To directly quote an excerpt "Such large electron density changes can be accommodated due to the high double-layer capacitance of Au nanoparticles in water, the magnitude of which is two orders greater than in vacuum. Possibly, the magnitude of the double layer capacitance is the factor limiting the maximum cathodic polarization achievable in the event the hole scavenging and photoexcitation conditions are ideal."

(ii) In the absence of an electron acceptor, the accumulated electrons will slowly react with protons at the interface, which are formed by the hole-driven oxidation of the alcohol. This phenomenon was explicitly demonstrated in ref. 27 in the form of H₂ evolution as an alternative pathway to accumulated electron capture by the electron acceptor. This was further leveraged in <https://pubs.acs.org/doi/10.1021/acsenergylett.9b01688> where it was shown that pH of the medium controls the relative contribution of this pathway. This side reaction is yet another factor that limits electron buildup.

Given the recognition of these factors, it is incorrect to state that prior models allow for infinite charge accumulation.

We are pleased to read that the reviewer is satisfied with our answers.

Regarding the microkinetic rate expressions, we want to clarify that we did not state that the authors of reference 22 and 27 have not thought about the limitations of their model. We are aware they did. Rather the statement in our article is that the microkinetic model developed in said references is incomplete, as it does not itself take into account the effects that could limit the charging. The microkinetic model itself therefore does mathematically predict an unlimited charging of the particle. This remains the weak point of this model and justifies the introduction of a new point of view as done in our manuscript.

We agree that the previous version of the manuscript may give the impression that the authors of the microkinetic model are unaware of its limitation. This was not our intention. We have therefore added some clarifying sentences to the revised version of the manuscript.

In contrast, the *microkinetic* model limits the accumulation of charge by the relative rates of oxidation and reduction. As a result, in the case that only one of the processes occurs, the model predicts the infinite accumulation of charge carriers on the particle. It is important to note that the authors of this model were aware of this shortcoming and proposed that Coulomb repulsion is probably a limiting factor to the achievable charge accumulation.²² As they did, however, not make an attempt to include the Coulomb repulsion into their model, its impact on the charging process remains largely unclear. To achieve a deeper understanding of the nanoparticle photocharging process, elucidating this aspect is therefore crucial.

Moreover, we also included a statement regarding the oxidation by protons (thank you for this remark):

The most likely oxidizers are the protons at the interface, which were formed by the hole-driven oxidation of the alcohol.²⁷ Moreover, also residual trace amounts of oxygen, and the water itself can play a role.²⁸

Reviewer #3 (Remarks to the Author):

I am fully satisfied with the answers provided by the Authors to my and other reviewer's reports. The paper can be published now.

We are pleased to read that the reviewer is satisfied with our answers.

Reviewer #4 (Remarks to the Author):

The authors have replied to my comments and also added material, in particular to the discussion related to the EtOH concentration dependence. Provided that the comments of the other reviewers have also been addressed satisfactorily, I can recommend the publication of the manuscript.

We are pleased to read that the reviewer is satisfied with our answers.

Reviewer report on the manuscript "In situ observation of nanoparticle photocharging: gold nanorods as photochemical capacitors" by F. Stete, M. Bargheer, and W. Koopman.

The Manuscript reports on the shift of the longitudinal plasmon of gold nanorod upon the photochemical reaction. Thus, the light induced reaction can be followed in real time. The authors attribute the shift of the plasmon resonance energy to the photoinduced charging of the gold nanorod, so that this is the charging process that they pretend to observe in the real time.

For the reason given below I recommend rejection of the present work.

Main criticism.

Early works of Mulvaney (see e.g. ref 4, 6, 7 of the present manuscript) lead to an erroneous idea that the charging of metallic nanoparticles would lead to the shift of the plasmon resonance. In turn, the shift of the plasmon resonance can be used to trace the charging. Indeed, the proposal seems simple and attractive. E.g. for the small free-electron metal spherical nanoparticle, the nonretarded dipolar plasmon resonance frequency of the spherical nanoparticle is given by $\omega_{DP} = \frac{1}{\sqrt{3}}\omega_b$. Here $\omega_b = \sqrt{n}$ is the bulk plasmon frequency (n is the **bulk** electron density, and atomic units are used). Thus, from the change of ω_{DP} one deduces the change in the bulk electron density Δn , and therefore the charge of the nanoparticle $Q = V \Delta n$ (V stands for the nanoparticle volume).

So far so good as far as formal Math is involved. The problem actually comes from the Physics. Classical electrostatics or quantum theory of metals tell us the same: there cannot be any excess charge in the volume of metal nanoparticle. Indeed, by virtue of the Gauss theorem, the presence of the charge in the volume will produce an electric field which would move the electrons until this field is screened. In other words, **for the finite size object the charge is rejected from the volume and**

resides at the surface. This is the so called Faraday's Ice Pail Theorem.

As follows from the quantum time-dependent density-functional theory calculations performed in *Langmuir* **32**, 2829–2840 (2016) (<https://pubs.acs.org/doi/10.1021/acs.langmuir.6b00112>), the surface charging is not sufficient to induce any appreciable shift of the plasmon resonance, and in any case, this shift does not follow the simple trends predicted by the \sqrt{n} dependence.

Therefore, the discussion presented in the manuscript is wrong, and the shift of the plasmon resonance is by no means the measure of the change of the 3D electron density. Most probably it reflects the changes in the medium surrounding the nanoparticle.